# Temperature and Consolidation Sensing Near Drinking Water Wells Using Fiber Bragg Grating Sensors

**Sandra Drusová** [1,2,*], **R. Martijn Wagterveld** [1], **Karel J. Keesman** [1,3] and **Herman L. Offerhaus** [2]

1   Wetsus, European Centre of Excellence for Sustainable Water Technology, Oostergoweg 9, 8911 MA Leeuwarden, The Netherlands; martijn.wagterveld@wetsus.nl (R.M.W.); karel.keesman@wur.nl (K.J.K.)

2   Optical Sciences, University of Twente, Hallenweg 23, 7522 NH Enschede, The Netherlands; h.l.offerhaus@utwente.nl

3   Mathematical and Statistical Methods—Biometris, Wageningen University, P.O. Box 16, 6700 AA Wageningen, The Netherlands

*   Correspondence: s.drusova@utwente.nl

**Abstract:** Drinking water wells require continuous monitoring to prevent groundwater-related issues such as pollution, clogging and overdrafting. In this research, optical fibers with fiber Bragg grating sensors were placed in an aquifer to explore their potential use in long-term well monitoring. Fiber Bragg grating sensors were simultaneously sensitive to consolidation strain and temperature, and these two responses were separated by creating autoregressive consolidation models. Consolidation responses from these multiple sensors were rescaled to obtain pressure distribution along the depth. Pressure and temperature data showed impermeable soil layers and locations where groundwater accumulated. Time development of temperature along the fiber revealed oxidation of minerals and soil layers with varying permeability. Fiber Bragg grating sensors are useful tools to examine subsurface processes near wells and they can show the first signs of clogging.

**Keywords:** fiber Bragg grating; ARX model; temperature; consolidation; well monitoring; subsurface iron removal; clogging

## 1. Introduction

Groundwater, as the cleanest source for public drinking water supply, is extracted from aquifers through wells. Series of pumping tests are performed before a well starts long-term extraction. Pumping tests are the most suitable way to determine groundwater quality, aquifer properties and appropriate depth of the pump from single observation points [1]. During a pumping test, groundwater is extracted from the well at a controlled rate and water levels are observed in monitoring wells. The aquifer properties calculated from pumping tests are hydraulic conductivity, storage coefficient, and specific yield [2]. These values are used to estimate the radius of influence of the area affected by pumping [3].

All aforementioned parameters might change during a long-term operation, so drinking water extraction wells need to be continuously monitored. A decrease in the well yield and water quality is the first sign of well clogging [4]. Continuously decreasing water levels in monitoring wells indicate groundwater overdrafting [5]. Consequently, groundwater gets extracted from a larger radius than initially expected and water quality might degrade due to intrusion of polluted or saline water.

When overdrafting continues, it might result in permanent land subsidence [6]. The process causing land subsidence is extraction-related consolidation. Consolidation is defined as volumetric changes in soil due to pressure changes. Groundwater extraction lowers the pressure in soil which results in vertical compression (strain) in the aquifer [7]. When the extraction stops and water levels repeatedly do not recover, compression of the aquifer accumulates [8]. Long-term monitoring of the wells is important to prevent all groundwater-related issues.

Commonly monitored parameters in extraction well fields are flowrate of the wells, pressure (water levels), temperature, water conductivity, and water quality (chemical composition) [9]. Another important parameter which needs to be monitored is groundwater flow around the well, both its magnitude and direction. Groundwater flow can be measured directly in the monitoring wells using borehole flowmeters [10] and the spatial flow field near wells can be predicted using groundwater flow models. The limitation of regional groundwater flow models is a lack of field data for the model input [11]. One way the models can be improved is by using input data with a higher spatial resolution collected by fiber-optic sensors [12].

Promising fiber-optic sensors for groundwater monitoring are fiber Bragg grating (FBG) sensors because they can measure multiple parameters simultaneously and spatial resolution can be customized [13]. FBG sensors are used to measure, for example, temperature and pressure [14,15], and pH [16]. An FBG sensor consists of a refractive index modulation in the fiber core with a period $\Lambda$. The grating acts as a mirror only for the Bragg wavelength $\lambda_0$—a wavelength that matches the grating period:

$$\lambda_0 = 2n_{eff}\Lambda, \tag{1}$$

where $n_{eff}$ is the effective refractive index for the propagating light mode in the fiber. FBG sensors are simultaneously sensitive to temperature and mechanical strain. FBG sensors can measure multiple parameters if these are translated to strain through FBG packaging. Sensitivity to a certain parameter can also be customized by choice of packaging. FBG sensors have a wide range of applications in monitoring subsurface structures. They are employed to prevent landslides [17], monitor the stability of dams [18,19], or measure pressure and temperature in oil wells [20]. Groundwater-related FBG research is focused on measuring pressure and temperature at remediation sites [21] and subsidence caused by groundwater extraction [22,23].

In this paper, we explore the potential of FBG sensors for groundwater monitoring in a drinking water well field. The wells in the field are used for groundwater extraction and injection. A small volume of the extracted water is injected into the ground for subsurface iron removal [24], causing clogging in soil instead of the well screen. The injected water can be used as a thermal tracer for groundwater flow and allows us to identify clogged soil layers. FBG sensors were deployed for measuring temperature and consolidation strain and a new method of separating these effects is presented. FBG consolidation response was rescaled to show pressure distribution close to a well. The results from the FBG sensors are compared to reference sensors for temperature, pressure and flowrate.

## 2. Materials and Methods

### 2.1. Experimental Site and Sensor Installation

The location of this study was a drinking water extraction field in Hengelo, Gelderland, The Netherlands. There are 12 wells in the area extracting with a flowrate of 128 m$^3$/h, see Figure 1a for their location. Groundwater from this area is rich in iron. Dissolved iron ions cause an unwanted yellow colorization in the drinking water, so they need to be removed before the water is sent into the distribution network. To remove iron, a part of the extracted water is enriched with dissolved oxygen and is periodically injected back into the ground with half of the normal extraction rate. As a result, iron oxides and hydroxides precipitate in soil.

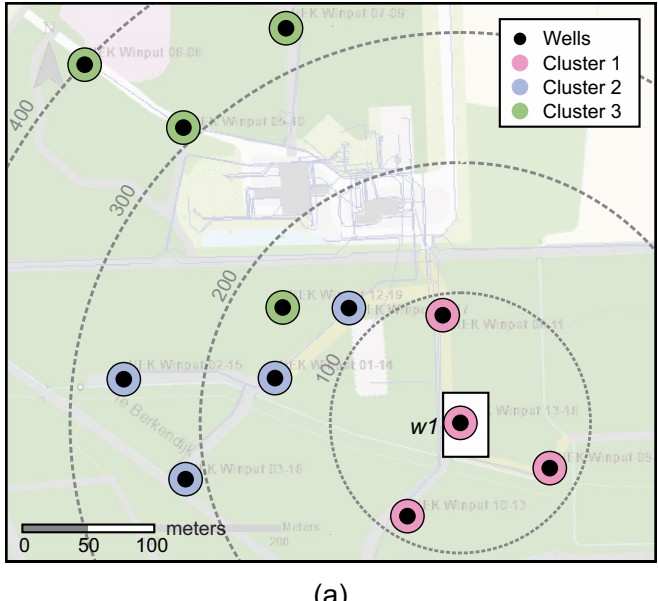
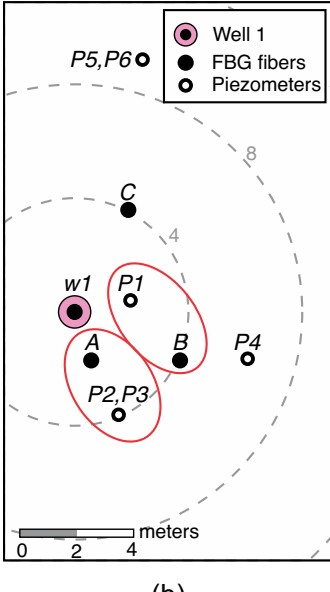

(a)                                                                          (b)

**Figure 1.** A map of the drinking water well field: (**a**) location of the extraction wells sorted in clusters. The area in the white rectangle is displayed in (**b**). (**b**) Location of the fiber Bragg grating (FBG) bundles and reference piezometers near well 1. Data from sensors circled in red is compared in the Section 3.

Data from multiple sensors are processed and compared in this paper:

- FBG sensors: consolidation and temperature
- reference divers: pressure and temperature
- weather station: atmospheric pressure
- flowmeter: horizontal flowrate in well $w1$
- status of 12 wells: on/off

FBG sensors and reference divers were installed at a distance of less than 10 m from well $w1$ (Figure 1b). Boreholes were drilled down to 38 m depth using the flush drilling technique. FBG fiber bundles with sensors were glued to a fishing line with a weight to keep the fibers straight when they were placed in open boreholes. The boreholes were refilled with the original sediment, although the natural sediment layering was not precisely reconstructed. Divers are pressure and temperature sensors that were suspended in piezometer tubes. Six divers (TD reference diver, van Essen Instruments, Delft, The Netherlands) were installed at the depth of 10 m.

In total, three FBG fiber bundles A, B, C were installed near the well, each bundle with 24 sensors equally spaced by 0.7 m, starting from 17 m depth to cover the length of the well screen (17.8–32 m). Each FBG bundle contains three fibers protected by a 1-mm thick Teflon tube with 3 mm diameter (Loptek, Berlin, Germany). The FBG bundle was a part of a sensing cable with DTS (distributed temperature sensing) fibers, a looped heating wire and a nylon fishing line. All parts were tied using tie wraps and glued with silicon.

FBG data was collected using an FBG interrogator (Hyperion si155, LUNA, Roanoke, VA, USA) controlled by a microcontroller (Raspberry Pi model 3B, Raspberry Pi Foundation, Cambridge, UK). The microcontroller was programmed to autonomously collect the data and report the state of the unit. FBG data had a sampling period of 2.1 s. Diver data had a sampling period of 5 s. Since divers were placed inside piezometers, corrections had to be made to get data comparable to the FBG sensors placed in direct contact with the soil. Divers measure absolute pressure $p_{diver}$, which is a combination of hydrostatic pressure $p$ and atmospheric pressure $p_{atm}$. Hydrostatic (pore pressure) was obtained as:

$$p = p_{diver} - p_{atm} \tag{2}$$

Atmospheric pressure was measured using a pressure sensor in a weather station installed at the site. It is important to realize that for most of the time, divers measured water temperature inside of the piezometer tube, not the groundwater temperature directly. The divers were suspended at the depth of 10 m, while piezometers screens were at 17.8–19.8 m and 22.5–24.5 m. When the nearest well starts extracting, groundwater flows through the piezometer screen but it does not reach the divers (see Figure 2a). When the water level in the piezometer drops, the diver measures the temperature of the water column above. This temperature tends to be higher due to stratification. When the nearest well stops extracting (see Figure 2b), the water level in the piezometer rises and the groundwater flows towards the diver. Inflowing water gets mixed with the water inside the piezometer and the temperature is equalized. Divers, therefore, measure the temperature of the groundwater in the aquifer only for a short time after the nearest well stops extracting.

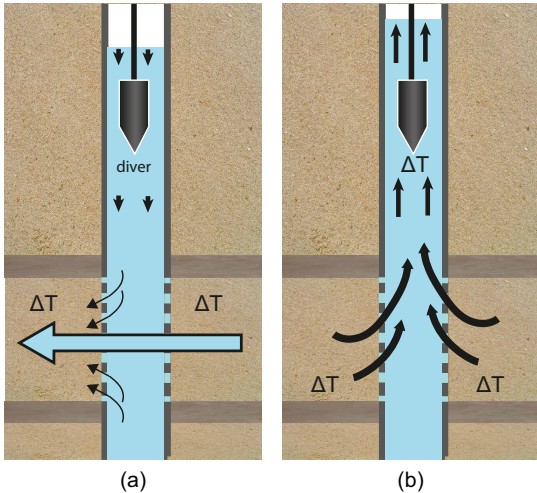

(a)    (b)

**Figure 2.** A diver installed in a piezometer several meters above the piezometer screen. (**a**) When the nearby well starts extracting, the extracted groundwater does not reach the diver. (**b**) When the nearby well stops extracting, groundwater flows up to the diver and the temperature change at that moment corresponds more closely to the groundwater temperature.

Horizontal flowrate in $w1$ was measured using a propeller flowmeter (van Essen Instruments, Delft, The Netherlands). The flowmeter was lowered inside the well during extraction and measurement was taken every 0.5 m, in the depths of the well screen. This measurement was taken shortly after the construction of the well in March 2000. All other presented data was taken in 2018–2019.

In Drusová et al. [25] we examined which effects cause a measurable response of the FBG sensors in this well field. It was discovered that the FBG sensors measure predominantly soil consolidation caused by groundwater extraction. FBG sensors are also sensitive to temperature changes. Pressure and drag force from the flow did not cause any FBG response. FBG response $\Delta\lambda$ in this well field has two components:

$$\Delta\lambda = \Delta\lambda_c + \Delta\lambda_T \tag{3}$$

where $\Delta\lambda_c$ is the consolidation strain contribution and $\Delta\lambda_T$ is the temperature contribution. We present a method to isolate each contribution and convert them to temperature and pressure.

### 2.2. Pressure Sensing

Groundwater temperature is quite stable with seasonal variations being lower than 0.5 °C. Unless water has been injected into the ground, the temperature contribution to the FBG response can be neglected:

$$\Delta\lambda = \Delta\lambda_c \tag{4}$$

The installed FBG sensors can measure pressure indirectly through consolidation. The FBG consolidation (strain) response depends on the pressure change $\Delta p$, the initial Bragg wavelength $\lambda_0$, the compressibility of the soil (represented as Young's modulus $E_{soil}$ and Poisson's ratio $\nu_{soil}$) and a friction coefficient $\beta$ between the FBG packaging and soil [25]:

$$\frac{\Delta \lambda_c}{\lambda_0} = \frac{(1 + \nu_{soil})(1 - 2\nu_{soil})}{(1 - \nu_{soil})E_{soil}}(1 - p_e)\beta \Delta p. \tag{5}$$

Consolidation strain from the soil is coupled to a wavelength shift through the strain-optic coefficient $p_e = 0.22$. The amplitude of the consolidation itself depends on the distance to a well.

The friction coefficient and soil compressibility are difficult to measure separately. However, if the pressure change $\Delta p$ is known, it is possible to calculate one constant scaling factor $F$ for each sensor which includes the effect of both soil compressibility and friction:

$$F = \frac{\Delta \lambda_c}{\lambda_0 \Delta p} = \frac{(1 + \nu_{soil})(1 - 2\nu_{soil})}{(1 - \nu_{soil})E_{soil}}(1 - p_e)\beta. \tag{6}$$

Scaling factors $F$ were calculated from a dataset where the nearest well $w1$ was not extracting. Pressure changes were caused by the extraction from further wells in the field. In this case, $\Delta p$ measured in all piezometers $P1 - P6$ was nearly equal even though the piezometer screens are located at different depths (12–37 m). Since the FBG bundles are located within 10 m distance of the piezometers at 17–33.1 m depth, it can be assumed that the $\Delta p$ measured by all FBG sensors is equal to the $\Delta p$ measured by the divers.

The $\Delta p$ from the divers was averaged and this curve was used to calculate the scaling factors $F$. The averaged $\Delta p$ and FBG curves were parsed into several intervals in such a way that $\Delta p$ is monotonically increasing or decreasing in the interval, see Figure 3. Since FBG sensors also measure temperature, parsing the FBG response into smaller intervals minimizes the temperature contribution because the temperature varies at much larger time scales. The scaling factor $F$ for each FBG sensor was calculated using Equation (6) in these intervals and averaged. Data from noisy FBG sensors was discarded. The sensors were considered noisy if their overall response to pressure changes caused by $w1$ was smaller than 2 pm (signal-to-noise ratio 2). The noise level of FBG sensors is dominated by the FBG interrogator, which has a noise level of 1 pm.

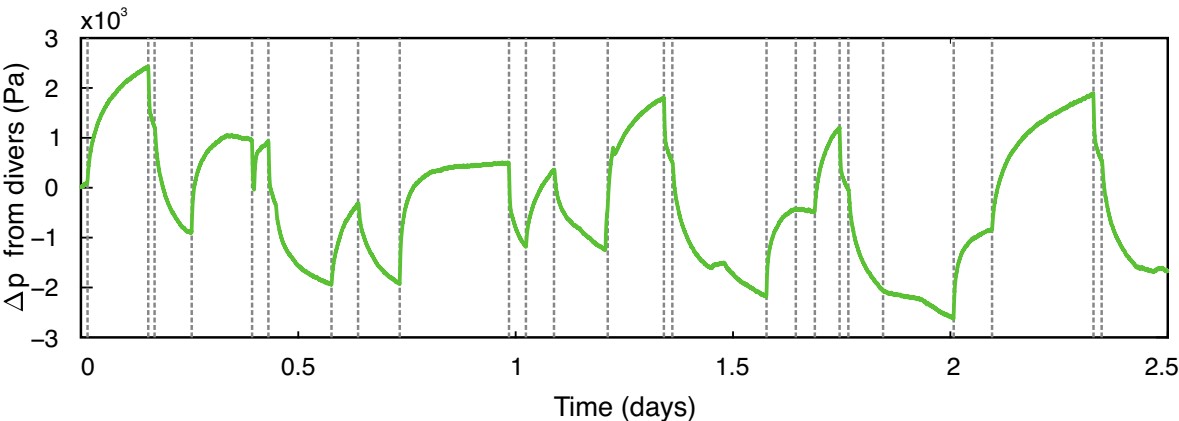

**Figure 3.** Average pressure change measured by all divers when $w1$ was not extracting. The pressure curve is parsed into several intervals divided by dashed lines where the scaling factor $F$ was calculated.

The scaling factor $F$ for each FBG sensor was then used to scale the measured data and calculate $\Delta p$ when $w1$ was extracting:

$$\Delta p = \frac{\Delta \lambda_c}{F \lambda_0} \tag{7}$$

The sensitivity of FBG sensors to pressure can be calculated from (7) as $\Delta\lambda_c/\Delta p$. The average pressure sensitivity at $\lambda_0 = 1550$ nm was $104$ pm atm$^{-1}$. FBG pressure data from multiple depths were compared to the pressure data from the reference divers (Figure 4).

$$\Delta\lambda_{\text{data}} - \underset{=0}{\Delta\lambda_T} = \Delta\lambda_c \longrightarrow \Delta p \ \text{(depth)} \ \text{---} \ FBG$$

$$\Delta p \ \text{(depth)} \ \text{---} \ divers$$

**Figure 4.** Flowchart showing how FBG pressure data is obtained and displayed in the results.

*2.3. Consolidation Models and Temperature Sensing*

The temperature in the aquifer is normally quite stable. Temperature changes occur during and after the injection of oxygen-rich water. The oxygen-rich water is stored in the reservoir in the basement of the pumping station and it travels through hundreds of meters of pipes before it reaches the aquifer. Due to heat exchange with the ground and the walls of the reservoir, oxygen-rich water has a higher temperature than the groundwater in the aquifer. Once the injection is finished, the groundwater temperature around a well is changed by heat convection due to extraction. The injected water can therefore be used as a tracer for the groundwater flow.

The FBG response during and after injection into $w1$ is a combination of consolidation strain and temperature, see Figure 5. The consolidation response is a variation on a scale of several hours, whereas the temperature response varies more slowly, on a scale of several days. These differences in time dynamics can be used to separate both effects. The consolidation response $\Delta\lambda_c$ can be predicted by a model as a response to the extraction wells. This predicted consolidation response can be then subtracted from the measured FBG response to recover the temperature variation $\Delta\lambda_T$:

$$\Delta\lambda_T = \Delta\lambda - \Delta\lambda_c \tag{8}$$

The model type used to analyze the FBG data is an ARX model. ARX model belongs to a general class of dynamic model structures and ARX stands for autoregressive with exogenous variables. This type of model was chosen because a relationship between extraction from wells and FBG response is unknown, so FBG response can be modeled from the measured data. ARX models are commonly used for groundwater level forecasting [26,27].

An ARX model is described by the following difference equation between time-dependent input $u(t)$ and output $y(t)$ [28]:

$$y(t) + a_1 \cdot y(t-1) + ... + a_{na} \cdot y(t-na) = b_1 \cdot u(t-nk) + ... + b_{nb} \cdot u(t-nb-nk+1) + e(t) \tag{9}$$

with the following terms:

$y(t), ... y(t-na)$—time shifted output values
$u(t-nk), ... u(t-nb-nk+1)$—time shifted input values
$e(t)$—error term
$a, b$—model coefficients
$na$—backward time shift in the output
$nb$—backward time shift in the input
$nk$—number of input samples that occur before the input affects the output, also called the dead time in the system.

Equation (9) can be simplified using a time shift operator $q$. The unit backward shift operator applied on a function $f(t)$ is written as:

$$q^{-1}f(t) = f(t-1). \tag{10}$$

The ARX model Equation (9) then becomes:

$$A(q)y(t) = B(q)u(t-nk) + e(t), \tag{11}$$

where

$A(q) = 1 + a_1 q^{-1} + a_2 q^{-2} + ... + a_{na} q^{-na}$ and
$B(q) = b_1 + b_2 q^{-1} + b_3 q^{-2} + ... + b_{nb} q^{-nb+1}$.

Thus $na$ and $nb$ determine the order of polynomials in $q^{-1}$. Dividing Equation (11) by $A(q)$ we can identify the transfer function $G(q)$ and the transfer function of the noise model $H(q)$:

$$y(t) = \frac{B(q)}{A(q)} u(t) + \frac{e(t)}{A(q)} = G(q)u(t) + H(q)e(t). \tag{12}$$

Commonly, the coefficients $a, b$ are found via least-squares estimation.

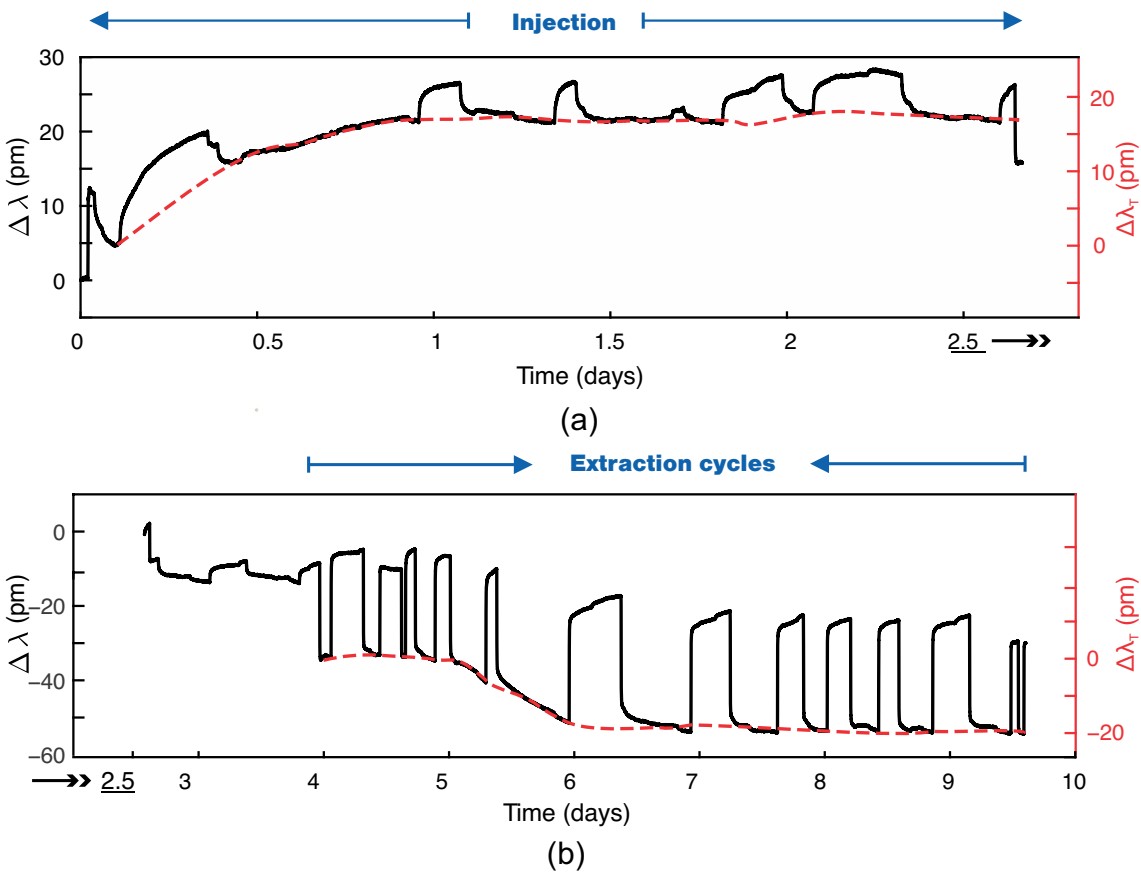

**Figure 5.** FBG response (consolidation strain and temperature): (**a**) during the injection, (**b**) after the injection of oxygen-rich water into $w1$. Temperature response is the baseline variation, consolidation response are the peaks. Temperature increased by 1.5 °C during injection and decreased by the same amount during extraction.

We assume that the FBG fibers placed in the soil behave as a linear time-invariant system. The FBG consolidation response is a sum of consolidation responses caused by the wells in the area. MISO (multiple input single output) ARX model has the following structure:

$$y(t) = G_1 u_1(t) + G_2 u_2(t) + ... + G_{NI} u_{NI}(t), \tag{13}$$

where $G_1, ..., G_{NI}$ are the transfer functions for each input up to the number of inputs $NI$. The consolidation response of every FBG sensor placed in the ground was modeled separately.

The output data for each model was the wavelength shift $\Delta\lambda$ of an FBG sensor and input data was the status of the 12 extraction wells. The status is a binary input represented as 0-off, 1-on, measured with a sampling period of 1 min. The FBG data was smoothed with a 2 min moving average and downsampled to 1 min period to match the pump data.

The optimal number of parameters for the models was chosen by comparing each estimated parameter value to its standard deviation to prevent overfitting. There were more $b$ coefficients assigned to the input from the nearest well to model this most dominant contribution more accurately. The time delay between inputs and output also varied. The input from the nearest well had no time delay and all other inputs were modeled with a delay of 1 sample (1 min). The delay corresponds to the time it takes to mobilize large volumes of water from a distance larger than 80 m. The number of parameters used for the model is shown in Table 1.

**Table 1.** The number of the autoregressive with exogenous variables (ARX) model parameters used for the FBG consolidation models.

|  | Output $\Delta\lambda$ | Input Well $w1$ | $w2 - 12$ |
|---|---|---|---|
| na | 6 | | |
| nb | | 8 | 2 |
| nk | | 0 | 1 |

In total, 24 ARX models were created for all FBG sensors at location A and 24 models for location B (see Figure 1b). The performance of the models was evaluated in terms of the root mean square error (RMSE) between modeled and measured data. The model requires training data not influenced by temperature. Temperature changes during the training and the validation period were measured by the divers and they were smaller than 0.1 °C, which translates into 1 pm of $\Delta\lambda$. 1 pm corresponds to the noise level of the interrogator so that the temperature contribution in the training and validation data was negligible. The length of the training dataset was 5 days and 7 days for the validation. Data for the training was chosen in such a way that all 12 input trajectories were unique to correctly estimate the contribution of each input to the FBG response.

Since the temperature changes obtained from $\Delta\lambda$ were expected to be smooth, a moving average filter was used on $\Delta\lambda_T$ to reduce errors caused by imperfect model predictions. Without any further strain effects, the temperature change $\Delta T$ can be calculated from $\Delta\lambda_T$ as [29]:

$$\Delta T = \frac{\Delta\lambda_T}{\alpha_T \lambda_0} \tag{14}$$

The value of the coefficient $\alpha_T = 7.8 \times 10^{-6}$ 1/°C was calculated from a thermal calibration experiment in a hot bath. The temperature sensitivity of FBG sensors can be calculated from (14) as $\Delta\lambda_T/\Delta T$. The FBG temperature sensitivity at $\lambda_0 = 1550$ nm was 12 pm °C$^{-1}$. The $\Delta T$ from 24 sensors at two locations (A, B) was linearly interpolated to display the temporal and spatial temperature variation near $w1$ after the injection of oxygen-rich water. FBG temperature data were compared to the flowrate $Q$ in the nearest well and temperature data from the reference divers (Figure 6). The compared FBG sensors and divers are placed in the same radial direction towards $w1$.

**Figure 6.** Flowchart showing how FBG temperature data is obtained and displayed in the results.

## 3. Results and Discussion

### 3.1. Pressure (FBG Sensors vs. Divers)

The majority of the FBG sensors at locations A and B measured a negative pressure change after the start of extraction from $w1$, see Figure 7. This response was expected. When groundwater gets extracted, the groundwater pressure decreases. The amplitude of pressure change measured by the FBG sensors is comparable to the nearby reference divers in piezometers $P1$ and $P2$. To be specific, only the maximum $\Delta p$ from the FBG sensors in this piezometer screen interval is comparable to the $\Delta p$ from the divers. Each piezometer tube has a screen that is 2 m long, so that a diver likely measures the largest pressure difference from a 2 m layer.

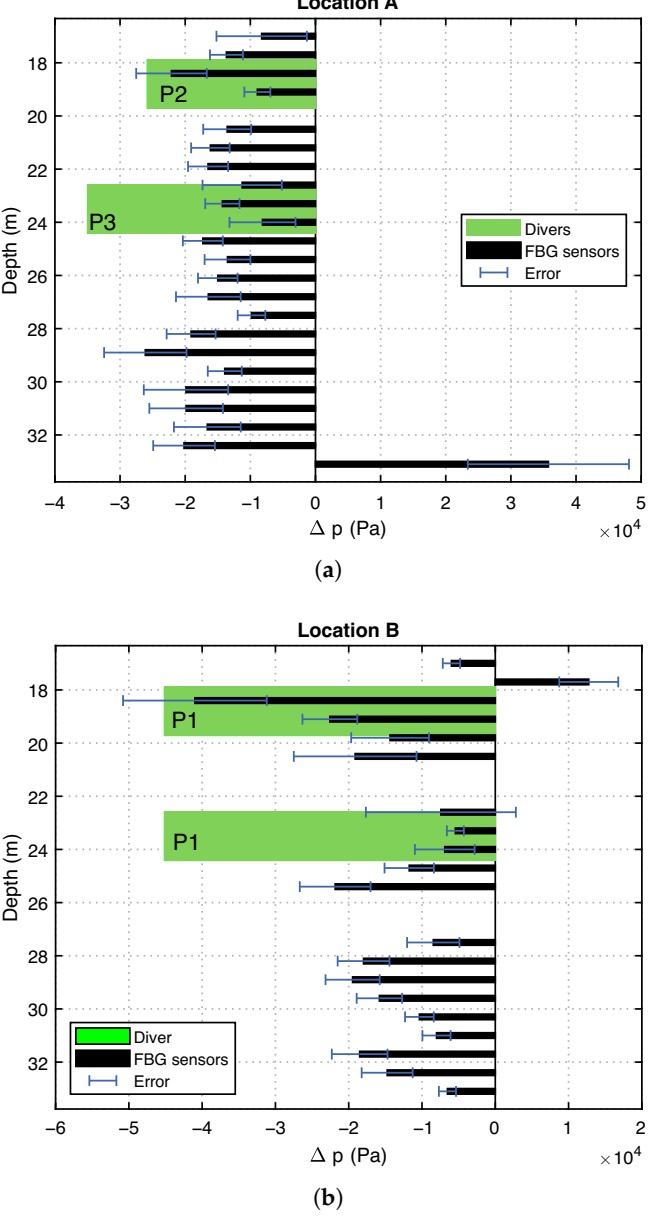

(**a**)

(**b**)

**Figure 7.** Groundwater pressure change after the start of the extraction from $w1$. (**a**) The FBG sensors at location A compared to two divers in the piezometers $P2$ and $P3$. (**b**) The FBG sensors at location B compared to a diver in the piezometer $P1$. Compared sensors are from the red-circled areas in Figure 1b. The piezometer tubes have screens in the displayed depth range, FBG are point sensors. Missing FBG data were noisy. The errors are a result of averaging during calculation of the scaling factors $F$.

The pressure data from the reference diver in *P*3 at 22.5–24.5 m shows a much larger change than measured by the FBG sensors (Figure 7a). The piezometer *P*3 is located further from *w*1 than the FBG bundle A, so this result seems counter-intuitive as pressure is expected to decrease with distance from the well. There are several possible explanations for this behavior. One possible explanation is that soil at *P*3 is more permeable. This could be caused by the presence of gravel or stones. When boreholes were refilled with the original sediment, the natural layering was not completely restored, and maybe gravel got concentrated in this layer. The natural layering in the boreholes should be restored in future experiments for more reliable results. The most likely explanation is that there is a high-pressure layer in between two FBG sensors that are 70 cm apart. The pressure difference from this layer is visible in the diver data because divers measure maximum $\Delta p$ from the piezometer screen depth.

There are two FBG sensors that show a pressure increase after the start of the extraction. They are at 33.1 m depth for location A and at 17.7 m depth for location B. The FBG sensor with a reversed response at location A is placed in a clay layer according to soil logs. The clay layer seems to swell during extraction and compress when the extraction stops. Detailed soil samples were not available for location B. The sign of the FBG pressure response can potentially be used to identify clay layers in the subsurface.

### 3.2. Consolidation Models

Consolidation models were created to separate the temperature response from the FBG data. This section discusses the accuracy of the models and its influence on the temperature calculation.

The FBG consolidation models were trained to an accuracy larger than 90% and had an accuracy larger than 75% for the validation data, see Table 2. The models are not accurate for several FBG sensors with a consolidation response $\Delta\lambda_c$ smaller than 3 pm, which is very close to the noise level of 1 pm. However, these noisy sensors could still be used for extracting $\Delta T$. Even though the consolidation response was small, the temperature response was still visible in the baseline variation and was obtained after the FBG response was post-processed with a moving average filter.

**Table 2.** Consolidation models for the FBG sensors at locations A and B: RMSE for the training and validation data. Values greater than 85% are in bold. Noisy sensors are underlined.

|   | Depth (m) | 17.0 | 17.7 | 18.4 | 19.1 | 19.8 | 20.5 | 21.2 | 21.9 |
|---|---|---|---|---|---|---|---|---|---|
| A | RMSE training (%) | **88.4** | **94.0** | **94.6** | **92.1** | <u>73.7</u> | **89.8** | **90.7** | **94.5** |
|   | RMSE validation (%) | **86.8** | **91.8** | **92.7** | **89.4** | <u>71.8</u> | **86.5** | **88.2** | **92.0** |
| B | RMSE training (%) | **92.7** | **91.2** | **96.7** | **96.5** | **90.9** | **91.4** | <u>**91.8**</u> | <u>35.1</u> |
|   | RMSE validation (%) | 80.9 | 75.1 | **89.2** | **86.7** | 79.5 | 75.8 | <u>−111.9</u> | <u>1.1</u> |

|   | Depth (m) | 22.6 | 23.3 | 24.0 | 24.7 | 25.4 | 26.1 | 26.8 | 27.5 |
|---|---|---|---|---|---|---|---|---|---|
| A | RMSE training (%) | **94.1** | **94.6** | **89.0** | **92.3** | **92.8** | **95.0** | **94.0** | **88.5** |
|   | RMSE validation (%) | **92.4** | **92.2** | **89.1** | **87.3** | **90.2** | **91.5** | **91.6** | **86.7** |
| B | RMSE training (%) | **89.5** | **89.6** | **86.7** | **93.2** | **96.0** | <u>35.4</u> | <u>69.4</u> | **85.5** |
|   | RMSE validation (%) | **87.0** | 72.8 | 76.3 | 78.0 | **91.5** | <u>0.7</u> | <u>22.8</u> | 76.6 |

|   | Depth (m) | 28.2 | 28.9 | 29.6 | 30.3 | 31.0 | 31.7 | 32.4 | 33.1 |
|---|---|---|---|---|---|---|---|---|---|
| A | RMSE training (%) | **95.1** | **96.4** | **94.7** | **89.4** | **92.7** | **87.7** | **92.1** | **89.5** |
|   | RMSE validation (%) | **93.5** | **92.5** | **93.0** | **90.4** | **89.8** | 82.1 | **88.5** | **86.3** |
| B | RMSE training (%) | **96.3** | **96.5** | **95.6** | **94.4** | **90.0** | **94.0** | **90.1** | **87.7** |
|   | RMSE validation (%) | **93.2** | **93.0** | **91.9** | **89.1** | 77.2 | 80.6 | 71.9 | 80.82 |

The consolidation response of the majority of the installed FBG sensors is a rescaled version of the pressure response measured by the divers. Figure 8 shows an example of an FBG sensor that behaves differently due to decoupling of the soil–packaging interface. The ARX model can also capture this behavior because it has an autoregressive part, as long as this behavior is consistent. The amplitude of the consolidation response did not change during a three-month-long experiment. Any changes could

have indicated soil clogging or deterioration of FBG packaging, but three months is most likely a short period of time to observe any of these effects.

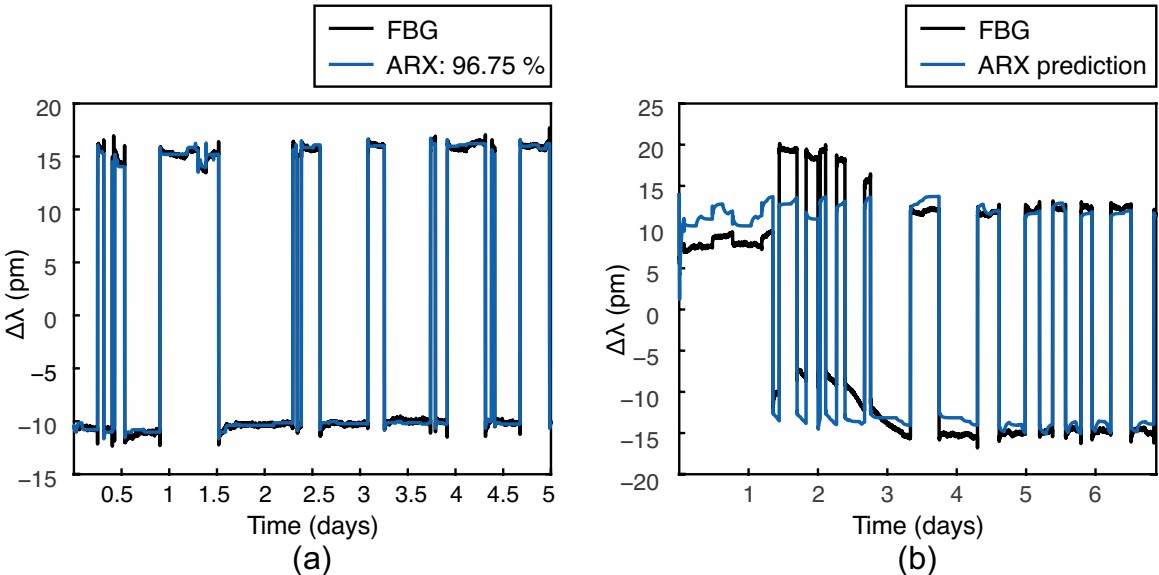

**Figure 8.** Comparison of the measured with modeled FBG data. The displayed FBG sensor is at 18.4 m depth at location B. (**a**) ARX model training and model accuracy, (**b**) Prediction using the ARX model after the injection of oxygen-rich water.

### 3.3. Temperature (FBG Sensors)

For both locations A and B, temperature changes are visible only after $w1$ starts extracting again, which is at t = 1.4 d after injection, see Figure 5b. This means that visible temperature changes are caused by heat convection with extracted groundwater. Groundwater and soil near the well warm up during the injection of oxygen-rich water (Figure 5a) and temperature starts decreasing when colder water from the aquifer is drawn towards the well. The rate of temperature decrease can therefore be associated with the magnitude of groundwater flow. Layers with varying permeability can be identified in Figure 9 based on temperature. The sensors that cool down faster are likely in a more permeable layer with higher flow. More permeable layers are shown as blue/turquoise and less permeable layers are green/yellow in Figure 9.

To quantify the groundwater flow it is necessary to know also the temperature difference between the injected and extracted water. This is not possible with the presented method. First, it was assumed that all FBG sensors in a fiber bundle have the same absolute temperature after the injection. This assumption is based on the observation that absolute temperature measured by the divers in the piezometers $P2 - P3$ was 10.6 °C and 10.47 °C. The reference piezometers have screens at different depths, so temperature along the fiber bundle most likely does not vary considerably after injection. The same applies to regular extraction—the temperature measured in the piezometers $P2 - P3$ shortly before the injection was 9.4 °C in both. If our initial assumption is true, this means that the thermal conductivity coefficient is constant within the well screen depth. The well screen area contains sand of grain sizes 0.3–0.5 mm and, according to experiments by Smits et al. [30], grain size has almost no effect on the thermal conductivity coefficient. FBG sensors can therefore detect relative differences in horizontal groundwater flow as a function of depth.

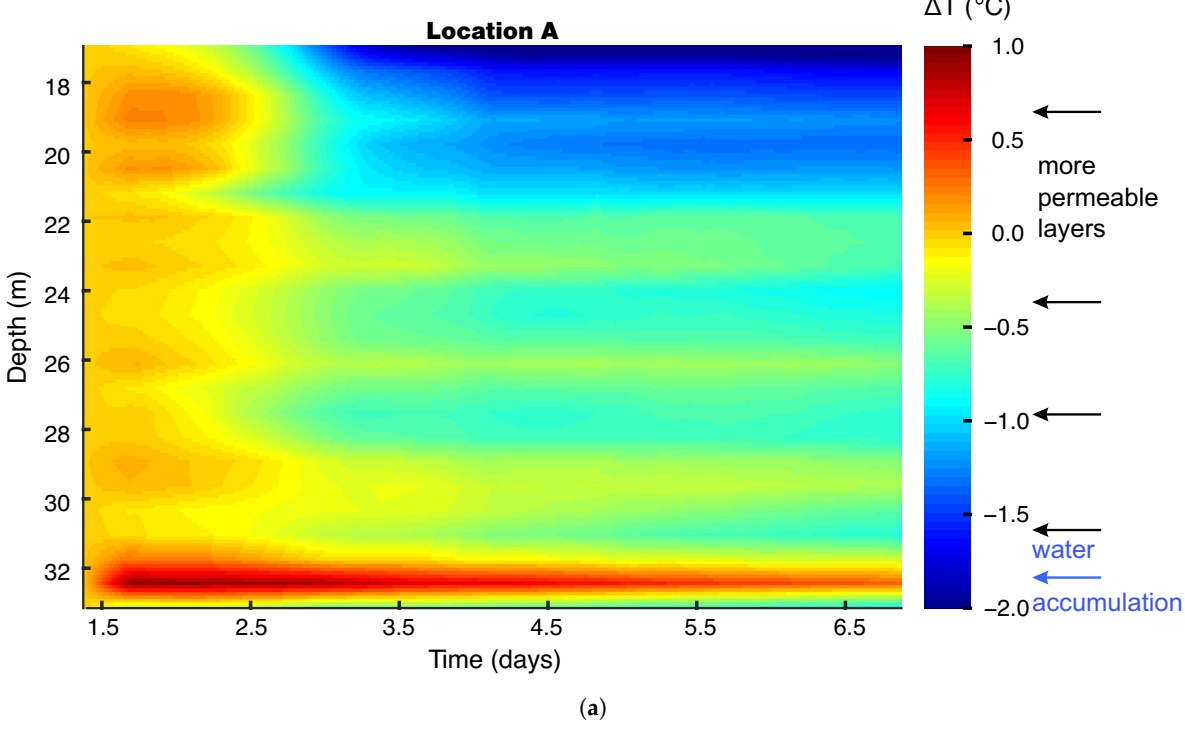

(a)

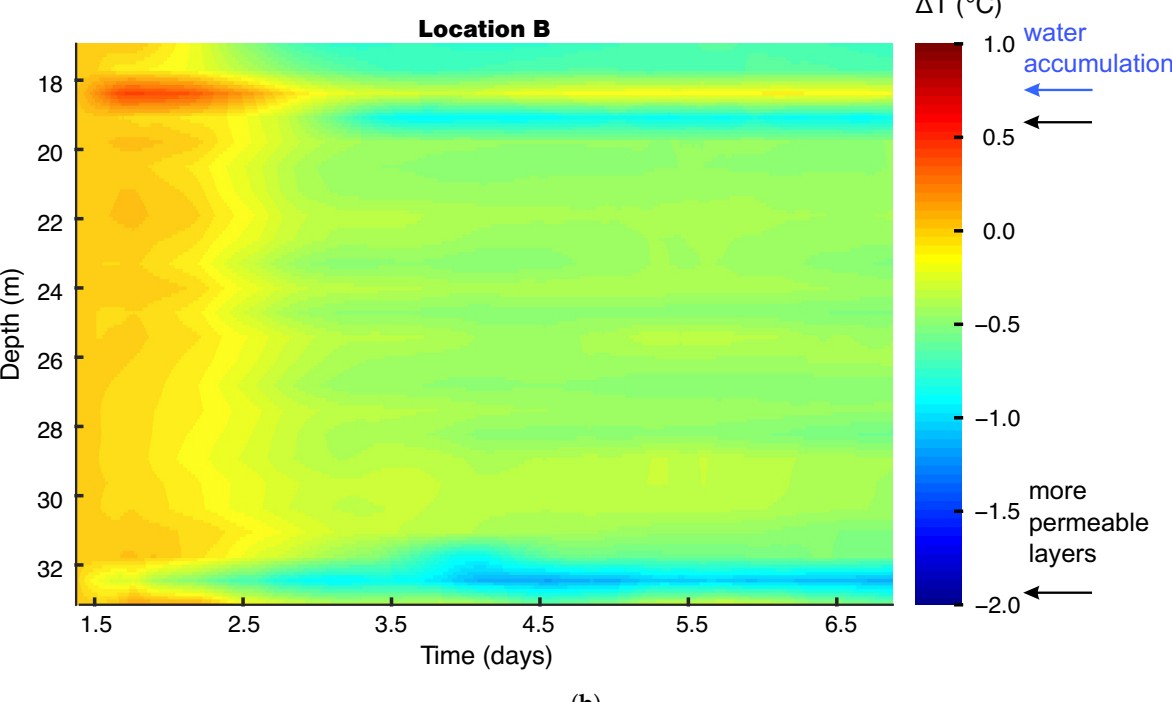

(b)

**Figure 9.** Temperature difference measured by the FBG sensors at (**a**) location A, (**b**) location B, after the injection of oxygen-rich water to $w1$. The injection finished at time $t = 0$. The data starts at 1.4 d after the injection, right when $w1$ started extracting again. All FBG sensors in each fiber bundle were assumed to have an equal absolute temperature at $t = 1.4$ d. Temperature data was linearly interpolated.

Both Figure 9a,b show one layer where temperature first increases at the start of the extraction and then decreases. For location A, this layer is located at 31.5–32.5 m and for location B at 18–19 m. These layers are close to the sensors with a reversed pressure response, as seen from Figure 7 (33.1 m depth at location A, 17.7 m depth at location B). The FBG sensors with reversed pressure response

indicate the presence of clay that may be partially or completely blocking the groundwater flow. As a result, the groundwater may accumulate near the clay layer. This effect can also be seen in Figure 10 from a sensor placed above a clay layer. The response during the first extraction period at t = 1.4 d considerably differs from the following extractions, both in amplitude and shape. An expected consolidation response at the start of extraction is a negative wavelength shift showing compression of soil. The measured FBG response at t = 1.4 d shows an expansion instead, which is most likely caused by local accumulation of groundwater. Accumulation can also be linked to the temperature increase of 1 °C. Since the accumulated water is rich in dissolved oxygen, ions can form by oxidation of minerals which is accompanied by heat release [31,32]. This exothermic reaction might be the cause of the observed temperature increase in the FBG data. Iron potassium phyllosilicate mineral glauconite was present in soil according to soil logs.

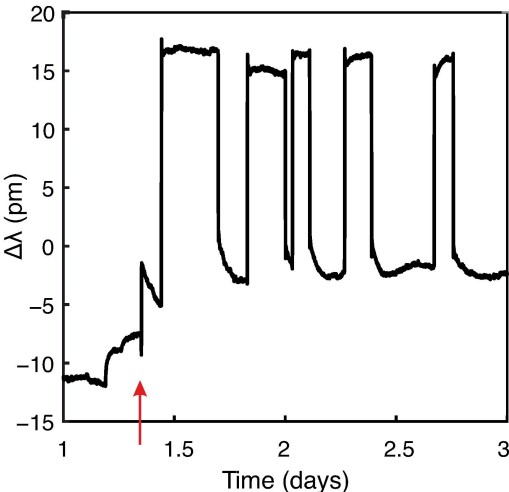

**Figure 10.** Response of an FBG sensor at 31.7 m depth at location A. The red arrow points at the start of the extraction in *w*1 after the injection of oxygen-rich water finished. At this moment the sensor shows soil expansion.

### *3.4. Temperature (FBG Sensors vs. Divers)*

The FBG temperature was compared to the temperature measured by the reference divers to assess if the $\Delta T$ calculation method suggested in this paper gives realistic results. For three cases presented in Figure 11, temperature curves measured by both types of sensors had a similar shape, and differences in the amplitudes were smaller than 0.2 °C. These differences are realistic considering the fact that the FBG bundles and corresponding divers were not placed at identical locations. Small oscillations in the FBG curves are caused by imperfect predictions of the FBG consolidation model. The FBG sensors can be used as temperature sensors in the soil, offering a much higher spatial and temporal resolution than divers. Even though the natural soil layering was not completely restored when the boreholes were filled, the results from the FBG sensors and divers are still comparable. Boreholes are not expected to cause considerable changes to the overall groundwater flow in the aquifer.

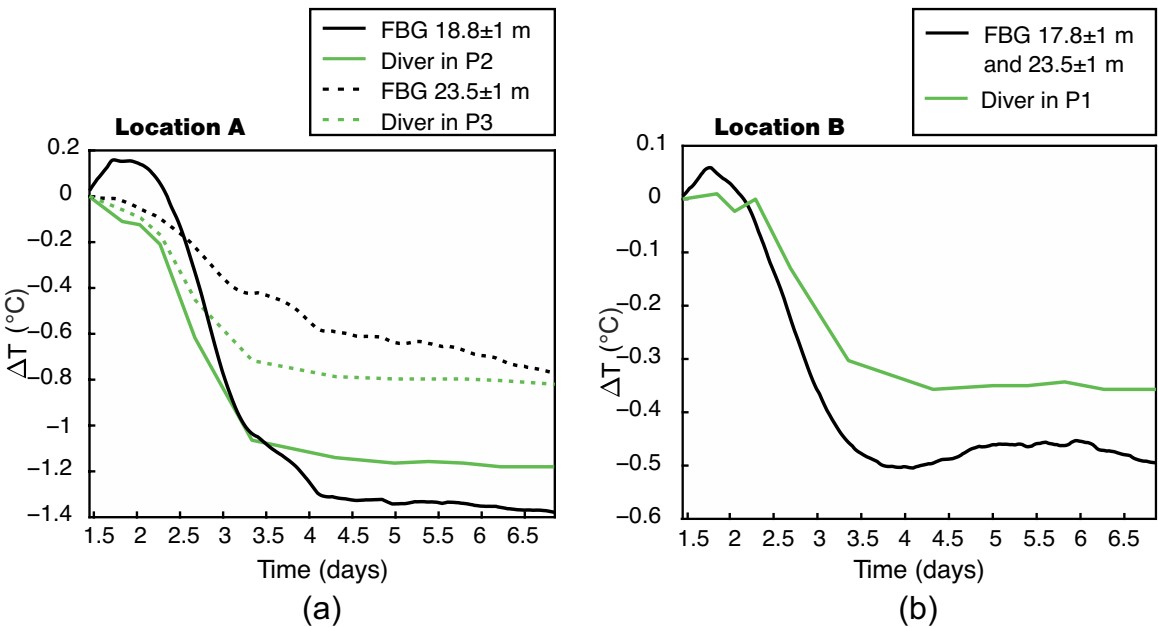

**Figure 11.** Temperature measured by the FBG sensors and the reference divers from the depth of piezometer screens. (**a**) FBG sensors at location A and the divers in piezometers *P*2 and *P*3. (**b**) FBG sensors at location B and the diver in piezometer *P*1. Compared sensors are from the red-circled areas in Figure 1b.

### 3.5. Temperature (FBG Sensors vs. Well Flowrate)

The FBG temperature data was compared to the horizontal flowrate measured inside the well *w*1 in Figure 12. Layers with varying permeability at location A are also visible in the flowrate data. The most permeable layer according to flowrate data is clearly identified in Figure 12a around 17 m as the most rapid temperature decrease. Two less permeable layers with flowrate below 2 m$^3$/h are also visible in this figure, around 22 m and 30 m. The flowrate data and FBG data do not match at 31 m. There is a highly permeable layer with a flowrate of 14 m$^3$/h, but FBG data from this depth imply the contrary. However, the permeable layer at the bottom of the well screen is visible in Figure 12b. At location B, the depth of 19–31 m contains less permeable layers.

Based on these results, it looks like the groundwater flow velocity varies with direction and depth. Permeable layers are more visible at location A, which is 1.7 m far from the well. Therefore there is likely more flow in the south-south-east direction than south-east-east towards location B, see Figure 1b. Groundwater flow at the bottom of the well screen is almost as high as the top. A highly permeable layer at the bottom of the well screen is not visible in Figure 12. Lower permeability than expected means this layer does not currently contribute to the flowrate in the well. It is not clear whether this has always been the case or the soil got clogged with iron over time. The flowrate data were collected shortly after the construction of the well, and FBG data were collected 19 years later.

Subsurface iron removal favors certain soil layers due to preferred flow paths [24]. When specific layers are more permeable during the extraction, they are most likely also more permeable during the injection. Higher flow means that oxygen-rich water reaches further from the well. If we know the magnitude of groundwater flow near the well, it is possible to estimate the maximum distance from the well where iron oxide formation is expected. Pressure and temperature data measured by the FBG sensors can be used as inputs to hydrological models to quantify groundwater flow near wells. Periodically collected FBG data can show the development of soil clogging near wells.

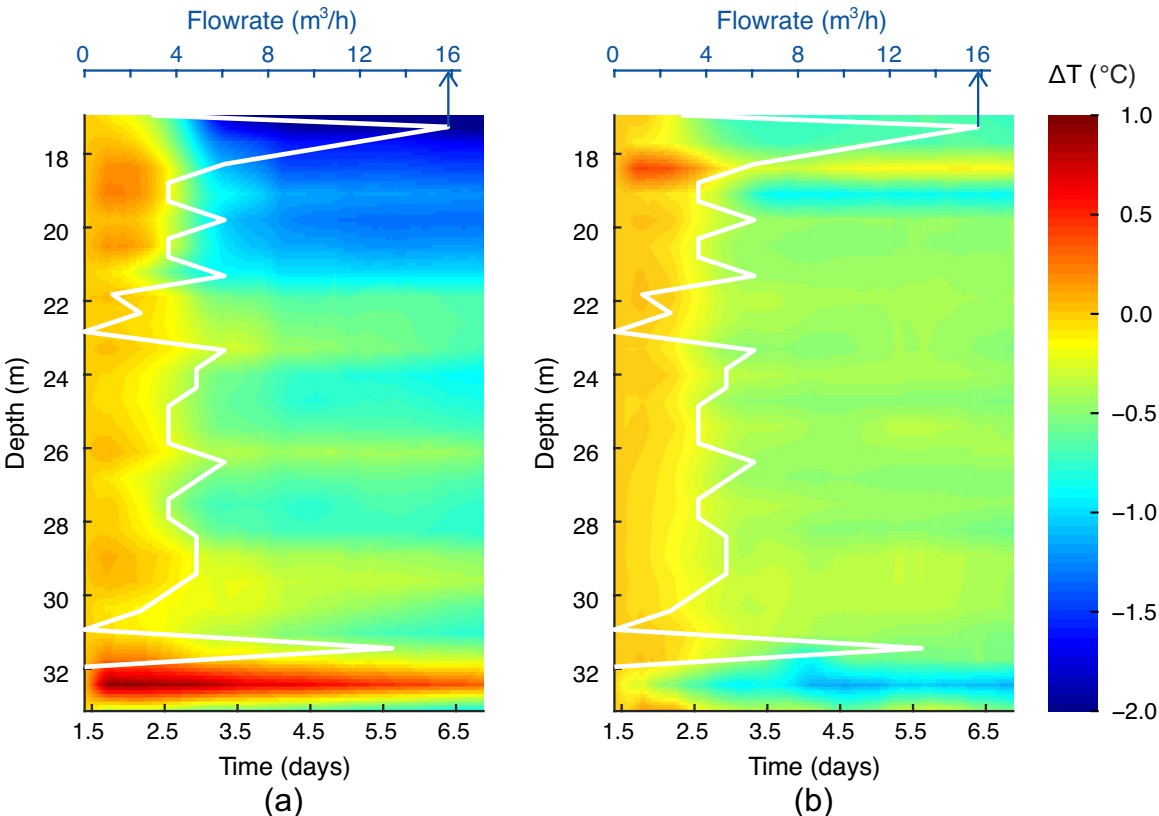

**Figure 12.** Temperature differenc measured by the FBG sensors at (**a**) location A, (**b**) location B, compared to the horizontal flowrate (white line) measured inside $w1$.

## 4. Conclusions

FBG sensors were placed near a drinking water well to investigate the possibilities for groundwater monitoring. The FBG sensors simultaneously measure consolidation strain and temperature. Consolidation strain is always measured as long as the well is in operation—extracting or injecting water, while temperature effects are visible only for a few days after injection. Therefore, it was possible to select FBG data that contained only a consolidation response and use that to train ARX models. The consolidation response predicted by the models was subtracted from the measured FBG data following injection to separate the temperature contribution.

The consolidation response is caused by changes in groundwater pressure. Extracting the pressure from the response is not straightforward because the friction coefficient between the soil and fibers and compressibility of the soil is unknown. However, the FBG sensors can be calibrated using the pressure measured by reference divers. Rescaled FBG consolidation data were used to calculate a pressure distribution near an extracting well.

The spatial distribution of pressure and temperature that was extracted from the FBG data can be used to study subsurface processes near wells. Water injected into the aquifer has a higher temperature than the groundwater, so it can be used as a tracer for groundwater flow. Soil layers with varying permeability were identified from the temperature data. The temperature data also showed layers where groundwater accumulates close to a clay layer. Clay layers were identified from the pressure distribution because FBG sensors in clay continuously show a reversed pressure response.

Compared to divers, FBG sensors offer higher spatial and temporal resolution and give more information about heterogeneous environments. ARX modeling can be used to decouple pressure and temperature and accurately predict these values. FBG consolidation, pressure, and temperature data can support groundwater flow models to quantify the flow near wells. FBG sensors are a helpful tool for long-term well monitoring and they can reveal the first signs of soil clogging.

**Author Contributions:** Conceptualization: S.D., R.M.W., H.L.O., methodology: S.D., R.M.W., H.L.O., software: S.D., validation: S.D., K.J.K., formal analysis: S.D., investigation: S.D., resources: S.D., data curation: S.D., writing—original draft preparation: S.D., writing—review and editing: S.D., R.M.W., K.J.K., H.L.O., visualization: S.D., supervision: R.M.W., H.L.O. All authors have read and agreed to the published version of the manuscript.

**Funding:** Wetsus is co-funded by the Dutch Ministry of Economic Affairs and Ministry of Infrastructure and Environment, the European Union Regional Development Fund, the Province of Fryslân, and the Northern Netherlands Provinces. This research received funding from the European Union's Horizon 2020 research and innovation programme under the Marie Skłodowska-Curie grant agreement No. 665874.

**Acknowledgments:** This work was performed in the cooperation framework of Wetsus, European Centre of Excellence for Sustainable Water Technology (www.wetsus.nl). We are grateful to the participants of the research theme "Groundwater technology" for fruitful discussions and financial support. The authors also thank Caspar V.C. Geelen for advice on data proceessing.

**Conflicts of Interest:** The authors declare no conflict of interest. The funders had a role in the decision to publish the results.

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
