# Peer review of "Temperature and Consolidation Sensing Near Drinking Water Wells Using Fiber Bragg Grating Sensors"

_water, doi:10.3390/w12123572_

Round 1

Reviewer 1 Report

This submission concerns the use of fibre Bragg gratings used to monitor temperature and ‘consolidation’ simultaneously in underground water extraction wells.    This reviewer’s perspective is from one who knows about the Bragg grating sensor, but is not at all familiar with the problem under consideration. 

At a general level the submission needed to be considerably shortened – aim for half the current length!  Also from a sensing perspective, the was really only one point of interest – the means whereby the ‘consolidation’ and temperature readings are separated.  The presentation of this particular aspect also needed more clarity – the ‘ARX’ concept needed clear explanation and – at a more basic level it would be very helpful, even necessary, to explain what ‘consolidation’ actually means in this context. 

The ideas though are of interest – and using several identical sensors to extract the spatial distributions of these two parameters could generate much enthusiasm since a similar approach could well be applied in other contexts.  However, at present from a sensing engineering perspective the submission is really (a) too long by half and (b) needs to significantly improve the clarity of the presentation. 

Consequently, the recommendation from this reviewer is that the current version be rejected and that the authors submit a completely rewritten version with these tow major points in mind.  However, from a water engineer perspective – the comments may be substantially different!

Author Response

We thank the reviewer for his/her positive comments and we would like to stress that this paper offers more than a method to separate two sensing parameters. The aim of the paper was not only to develop new FBG sensing methods, but also to show how hydrogeologists can benefit from FBG data compared to traditional sensors. Indeed, if this paper was written only from the sensing perspective, it would be half of the current length. We believe that the extra information obtained from FBG data is interesting for hydrogeology-oriented readers of the Water journal.

Several sections of the paper were rewritten to improve the clarity of presentation:

  • Fig. 4 and 6 were added to explain how FBG data was obtained and displayed in the Results section
  • The Temperature section in the Results and discussion was split into three sections based on Fig. 6
  • The section about soil clogging was deleted from the Results section as it did not focus on the FBG sensors
  • The concept of consolidation was further explained in the Introduction (second paragraph) together with relevant references for more information
  • The ARX models are commonly used in hydrogeology from groundwater level forecasting based on precipitation, temperature and water extraction. Relevant references were added to section 2.3 above eq. (9), also explaining why this type of model was chosen for data processing. The section 2.3 explaining the basics of ARX modeling was reorganized.
  • The current length of the paper is shorter by 2 pages

Reviewer 2 Report

General Comments: The author provided a follow-up and validation research work with the FBG sensors, based on the long-term well monitoring of the temperature and consolidation strain. As it was mentioned in his previous publication, this work also decoupled the FBG wavelength responses thorough establishing the autoregression models. The practical sensing test with FBG and model training was generally based on a single well. The sensing method has been validated in real situations and the trained model also demonstrated good accuracy. This paper could be accepted after the revision.  

Specific Comments:

  1. In Figures 4a and 4b, the temperature between the injection and extraction was not given. It seems that the resonance wavelength has undergone abrupt changes in this process: the wavelength of 4a has changed in the positive range, while that of 4b moved to a negative region.
  2. In this paper, the sensitivity and the standard error level of the FBG sensor was not given, which would be also important for readers to understand the sensing performance.
  3. According to Figure 8, the correlation between the FBG sensor and Divers was not quite good. Could the author further describe the difference between the two source signals, like what made the differences?
  4. As the author mentioned, the missing FBG data points showed low signal-to-noise-ratios. So, what are the standard SNR and acceptable SNR in this study? The noise should also be plot as error-bars in Figure 8.
  5. How did the author identify the location and depth of the clay and permeable layers?
  6. The clear structure and design of the proposed FBG sensor are not given in this article. In addition to temperature and strain, whether there are other factors that also affect the signal drift.

Round 2

Reviewer 1 Report

The authors have responded constructively to the previous comments and the submission is now fit for publication.